# Economic and Organizational Impact of COVID-19 on Colombia's Tourism Sector

Luz Natalia Tobón Perilla [1] , Elena Urquía Grande [1] and Elisa Isabel Cano Montero [2],*

[1] Faculty of Economics and Business, Universidad Complutense de Madrid, 28040 Madrid, Spain
[2] Faculty of Economics and Business in Talavera de la Reina, Universidad Castilla La Mancha, 45600 Talavera, Spain
* Correspondence: elisaisabel.cano@uclm.es

**Abstract:** The global COVID-19 crisis has strongly affected tourism. In an emerging economy like Colombia's, however, the pandemic's effects may differ from those experienced in more advanced countries. Building on prior studies, this investigation aims to determine the economic and organizational impact of COVID-19 on the tourism sector in the areas of lodging, travel agencies, clubs, and restaurants by identifying indicators relevant to the business tourism sector. We contrast data obtained empirically from a survey administered to a sample of 289 Colombian tourism SMEs. The model, developed with structural equations, enables identification of the factors with the greatest influence. The results indicate a high impact on sales and personnel expenses, leading to a decrease in management and innovation capability. In the gradual recovery process, internal measures taken by business owners to face the crisis have been more effective than measures taken by the government. Moreover, firms have prioritized financial strategies and innovation in marketing and services.

**Keywords:** COVID-19; strategies; structural equations; economic and organizational impact; tourism

## 1. Introduction

In an emerging economy with deficiencies in its companies' competitiveness and a low percentage of exports, the global COVID-19 crisis may affect economic development more strongly than in more advanced countries [1]. Figures from the World Bank and the International Labor Organization agree that tourism is the sector most affected globally, with a decrease in global GDP of approximately USD 2.4 billion and job losses of up to 80% [2,3]. Due to this impact, tourism has come to be considered a precursor of economic development in many countries.

Colombia's situation in early 2020 differed from that of other Latin American countries and intensified in April and May 2020. Reductions in gross revenue reached 63.4% in Latin American firms. The United Nations [4,5] reports that tourism production in Colombia fell USD 6.79–1.62 and the sector's participation in the nation's GDP decreased from 5% in 2019 to 2% in 2020.

To tackle the pandemic's negative effects on the tourism-based economy, we analyze the causal relationships identified by several authors [6–8], including business situation or positioning prior to the COVID-19 crisis and its influence on business management [9,10]. We also include organizational strategic management—specifically, orientation to establishing financial goals and strategies, relationship to customers, and monitoring of organizational objectives and their contribution to enduring the crisis [11–13]. Further, our review of business studies indicates, as relevant factors, processes of technology-based innovation and development to manage the economic and financial effects of COVID-19 [14–16]. However, none of the studies published to date contrast either all the variables proposed as a whole or the research on factor correlation.

The goal of this study is to determine the economic and organizational impact of the COVID-19 crisis on Colombia's tourism in the subsectors of lodging, travel agencies,

food and beverage, and tourism clubs to explain how significant business situations and organizational strategies are in facing crises. Our methodological design includes variables to measure the economic impact of the COVID-19 crisis, such as revenue variations, investment in reactivation, payroll recovery, remote work, amount of public aid, and ongoing impact up to December 2021.

We administered a structured survey to a stratified sample of 289 small- and medium-sized enterprises (SMEs) dedicated to tourism in Colombia, based on firms obtained from the Orbis database. Structural equation modeling was used to analyze the responses and test the causal effects among the variables for business situations when facing the impact of the COVID-19 crisis and the organizational strategic management and investment in technological innovation development. The variables were analyzed based on 25 factors linked to five hypotheses. Since this approach sought to confirm the strength of the relationships among the variables analyzed, we applied confirmatory factor analysis (CFA).

The statistical results confirm four of the five hypotheses proposed. The findings indicate a high impact on sales and number of workers in 2020. These two measures were linked most closely to the crisis and the business situation at the end of 2021, when recovery was still underway. The results also showed that financial management of firms that performed strategic management was less severely impacted than financial management of firms that did not. Further, although investments in innovation and technology decreased initially, these factors were subsequently strategic for supporting firms' reactivation and have become the most significant source of recovery, even beyond public aid.

This article is organized as follows. Next, we perform a theoretical review of the most important research antecedents. We then describe the methodological design, from data collection to analytical procedure and presentation of the results. Subsequently, we discuss the results and contrast them with similar studies. Finally, we present the conclusions.

## 2. Literature Review

### 2.1. Economic Impact of COVID-19 on SMEs and Tourism

The effects of COVID-19 on the tourism industry are evident in the decrease in approximately USD 2.4 billion in the sector's global GDP [2]. In Latin America, tourism represented 10% of exports in goods and services [4]. In Colombia, tourism's contribution to GDP decreased from 5% in 2019 to 2% in 2020. Expected revenue from the global tourism sector decreased from USD 712 to 396 billion and from USD 6.79 to 1.62 billion in Colombia. The number of international tourists fell globally by 73% in 2020, while the number arriving in Colombia fell 70% [5]. Job losses, thus, reached 80% [3].

The pandemic's main consequence for tourism is decreased demand for services, due to perception of risk and reduction in purchasing capacity [17–20]. In Colombia, this situation led to a 63% decrease in hotel revenue in November 2020 [21], which affected employment and productivity [22], as tourism is a significant source of economic development in many countries [23,24]. Tourism is also considered the sector most severely affected by the COVID-19 pandemic [25–29].

The pandemic's devastating effects on global tourism [30] have led to studies with varied approaches that focus on employment [31–33], human resources [34], fall in prices [35,36], and decrease in consumption and reserves [20,37–40]. Other topics researched include decline in revenue [41,42], decrease in profits per share [43] or profitability [44], and disadvantage to SMEs and less solvent firms [45,46].

Still, other studies explore public support to mitigate the crisis [47–51], finance and marketing strategy [52] support based on strategic groups [53,54], corporate social responsibility [55–58] technological innovation [59], leadership styles [60,61], and learning and knowledge transfer [62,63]. All these studies have contributed to understanding of the crisis caused by the COVID-19 pandemic.

## 2.2. Business Situation

Although the business situation caused by the pandemic has been studied from the perspective of revenue and number of workers [10,64], factors related to the pandemic's influence on the economic activities of accommodation, travel agencies, and food and beverage outlets are also relevant to calculating economic effects. Studying the impact on each activity enables us to understand whether business size, type of client, or other factors influence the way companies face different crises and the level of this impact on their economy [65,66].

Previous studies on business situation and the COVID-19 crisis have shown that the variables hotel size and infrastructure [67], customers and competition [68], SME revenue according to size [10], and declining payrolls are relevant factors in mitigating the economic impact of the pandemic [41]. Like ours, these studies were based on surveys and apply structural equations.

## 2.3. Strategic Management

Some studies of strategic organizational management [69,70] incorporate variables, such as planning and management for crisis recovery. The authors of [71] evaluated variables, such as development of policies and reformulation of strategies to reorganize tourism firms facing the effects of the crisis. The authors of [72] analyzed the relationship between the strengths and weaknesses of organizational performance during the period of the pandemic's greatest impact and found increased weakness in management.

Refs. [12,73], in contrast, studied the role of variables on strategies for managing financial performance (measured by profitability), liquidity, and debt–capital ratio in economic recovery of tourism companies and demonstrated the importance of this management. Ref. [74] studied liquidity risk management and financial flexibility as fundamental factors in times of crisis. Other financial strategies studied were deferral of capital payments, reduction in market expenditure to recover liquid assets [75,76], financial restructuring, and new sources of financing.

## 2.4. Innovation and Development

Among the innovative strategies firms implemented to cope with the COVID-19 pandemic, refs. [77,78] identified factors, such as differentiation of products and channels in the digital market (including social media). Their studies demonstrate the effectiveness of these measures. Other indicators have also been used to measure advances in digitalization and use of innovative knowledge as strategies to adapt to changes caused by the crisis. Digitization is the measure most recommended [79,80].

Other studies evaluated the consideration of innovation in business models as a measure to mitigate the effects of the COVID-19 crisis [15,81]. Additional research studying resilience and investment in reactivation demonstrates that technology and innovation capacity contribute to sustainability in tourism SMEs [16,59,82].

## 3. Materials and Methods

Based on this theoretical development, we propose the following hypothesis for the data analysis:

**Hypothesis 1:** *The business situation of tourism SMEs in Colombia may influence the economic indicators caused by the COVID-19 crisis.*

**Hypothesis 2:** *The COVID-19 crisis has affected the strategic management of tourism SMEs in Colombia.*

**Hypothesis 3:** *Innovation and development in tourism SMEs have been decisive, although conditioned by the COVID-19 crisis.*

**Hypothesis 4:** *Depending on the business situation of tourism SMEs, these firms promoted innovation and development, which contribute to improving the business situation.*

**Hypothesis 5:** *Innovation and development practices support organizational management of tourism SMEs.*

Figure 1 presents the theoretical approach and the relationships between the variables.

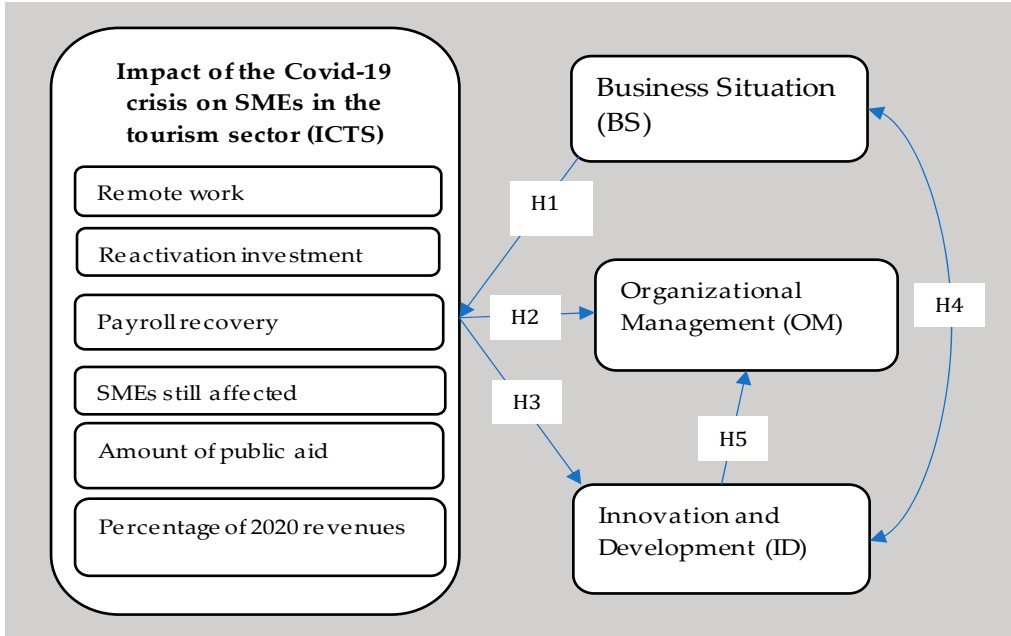

**Figure 1.** Theoretical model.

This study developed a structural equations model following following [60,82–86]. The model relates the variables for measuring economic impacts of the COVID-19 crisis on the business situation (and vice versa), organizational management strategies, and investment in innovation and technological development. The data were obtained from a survey (see Appendix A) [62,87,88] with responses measured on a Likert scale, following [88]. Sales and other financial data were extracted from the Orbis database.

To categorize the item "Tourism subsector" according to relative importance of each subsector, we assigned 1 to the subsector with the lowest representativeness or number of companies in the population and 4 to the subsector with the highest number of companies in the population. The items "Number of workers", "Sales volume", and "Productive capacity" were categorized by ranges adapted to a scale of 1 to 5, where 1 is greatly decreased and 5 is greatly increased. Appendix B presents the ranges and criteria for all the items' categorization.

The population of tourism firms in Colombia obtained from Orbis was 4766. Of this total, 1177 firms may be considered small or medium sized, following revenue criteria for firm size in Colombia, defined by Decree 957 of 2019. We used the following equation to calculate the sample:

$$n = \frac{N * Z^2 * p * q}{d^2 * (N-1) + Z^2 * p * q}$$

where:

N = population = 1177
Z = 95% confidence level = 1.96
p = expected probability of success = 0.5
q = probability of failure = 0.5

$d^2$ = precision (maximum admissible error) = 0.05

We, thus, obtained a sample size of 289, as shown in Table 1.

**Table 1.** Population and sample.

| Activity | Population | Sample |
|---|---|---|
| Lodging | 437 | 107 |
| Travel agencies | 186 | 46 |
| Recreation clubs and agrotourism | 69 | 17 |
| Prepared food and beverage | 485 | 119 |
| Totals | 1177 | 289 |

Note: Population and sample stratified by tourism subsectors.

The data were collected through an online form. Eighteen questionnaires were found to be incomplete or to have been completed by firms that did not belong to the tourism sector, leaving useful data from 271 SMEs. Sample size, calculated based on a 95% confidence level and 5% margin of error, showed this sample to be valid.

We applied CFA following [36,38] to test the theoretical constructs proposed for the causal relationship of COVID-19 pandemic impact to firms in the tourism sector. The analysis was based on these firms' business situation and the effects on organizational and economic–financial structure and innovation and development processes, as specified in Table 2. In this table, we added, in the fourth column, authors who have researched the different items and variables.

**Table 2.** Dimensions of the variables.

| Variable | Items | Factors | Authors |
|---|---|---|---|
| Business situation (BS) | Item 1 BS1 | Tourism subsector | Neise T., Verfurth P., Franz M. |
| | Item 2 BS2 | Number of workers | Melnyk S., Schoenherr T., Verter V. et al. |
| | Item 3 BS3 | Sales volume | Marjanski, A., Sulkowski, L. |
| | Item 4 BS4 | Main clients | Markovic S., Koporcic N., Arslanagic-Kalajdzic M. et al. |
| Organizational management (OM) | Item 5 OM1 | Formulation of income budget | Haqbin, A., Shojaei, P., Radmanesh, S. |
| | Item 6 OM2 | Expenditure budget | O'Toole C., McCann F., Lawless M. et al. |
| | Item 7 OM3 | Financial goals (type) | Ganlin P., Qamruzzaman M., Mehta A. et al. |
| | Item 9 OM4 | Cost identification | Hrivnák M., Moritz P., Chreneková M., |
| | Item 10 OM5 | Productive capacity | Doerr S., Erdem M., Franco G. et al. |
| Innovation and development (ID) | Item 11 ID1 | Investment in product development | Anggadwita, G., Martini E., Hendayani R., Adam N., Alarifi G. |
| | Item 12 ID2 | Marketing investment | Polas M., Raju V.—Rakshit S., Mondal S., Islam N. et al. |
| | Item 13 ID3 | Investment in process improvement | Rakshit S., Islam N., Mondal S. et al. |
| | Item 17 ID4 | Number of years (with I + D) | Yuniarty S. I., Abdinagoro S. et al. |
| Impact of the COVID-19 crisis on SMEs in the tourism sector (ICTS) | Item 18 ICTS1 | Remote work | Park S., Lee S., Cho J.—Bargados, A. Félix A. G. and García N.—Piga, C. A., Abrate G., Viglia G., and de Canio, F. |
| | Item 20 ICTS2 | Investment in reactivation | |
| | Item 21 ICTS3 | Payroll recovery (as of December 2021) | Chen C. F., Wang Z., Tang X. L. |
| | Item 22 ICTS4 | Still affected by the crisis | Le D. and Phi G. |
| | Item 24 ICTS5 | Amount of public support | Sanabria J. M., Aguiar T. and Araujo Y. |
| | Item 25 ICTS6 | Mehta K. and Sharma S. | |

Note: The first three variables are first-order latent variables. The fourth is the criterion variable (impact of COVID-19 on SMEs in the tourism sector). On the right side, the factors that compose each variable are defined, followed by the authors supporting the variables.

This set of constructs comprises the five hypotheses to be contrasted with the empirical data. They are H1: The business situation of tourism SMEs in Colombia may influence the economic indicators caused by the COVID-19 crisis. H2: The COVID-19 crisis has affected

the strategic management of tourism SMEs in Colombia. H3: Innovation and development in tourism SMEs have been decisive, although conditioned by the COVID-19 crisis. H4: Depending on the business situation of tourism SMEs, these firms promoted innovation and development, which contribute to improving the business situation. H5: Innovation and development practices support organizational management of tourism SMEs. Table 3 presents the descriptive statistics for the variables.

**Table 3.** Statistical data on the items.

| Items | Mean | Standard Deviation | Variance | Skewness | Kurtosis |
|---|---|---|---|---|---|
| Item 1 BS1: Tourism subsector | 2.5129 | 1.35223 | 1.829 | 0.022 | −1.81 |
| Item 2 BS2: Number of workers | 1.8561 | 1.11453 | 1.242 | 1.08 | 0.122 |
| Item 4 BS4: Main clients | 1.5391 | 0.58192 | 0.339 | 1.622 | 1.587 |
| Item 5 OM1: Formulation of income budget | 2.7269 | 1.22896 | 1.51 | 0.425 | −0.83 |
| Item 6 OM2: Expenditure budget | 2.6863 | 1.1962 | 1.431 | 0.442 | −0.761 |
| Item 7 OM3: Financial goals (type) | 2.8598 | 1.09995 | 1.21 | 0.449 | −0.428 |
| Item 9 OM4: Cost identification | 2.6384 | 1.13946 | 1.298 | 0.396 | −0.748 |
| Item 10 OM5: Productive capacity | 2.8081 | 1.08529 | 1.178 | 0.353 | −0.571 |
| Item 11 ID1: Investment in product development | 2.4465 | 1.20113 | 1.443 | 0.861 | −0.217 |
| Item 12 ID2: Marketing investment | 3.0037 | 1.05389 | 1.111 | 0.24 | −0.685 |
| Item 13 ID3: Investment in process improvement | 1.8856 | 0.68579 | 0.47 | 0.774 | 1.805 |
| Item 17 ID4: Number of years (with I + D) | 2.2657 | 1.27493 | 1.625 | 1.057 | 0.055 |
| Item 18 ICTS1: Remote work | 2.5018 | 0.93838 | 0.881 | 0.198 | −0.291 |
| Item 20 ICTS2: Investment in reactivation | 2.7565 | 1.10876 | 1.229 | 0.74 | −0.327 |
| Item 21 ICTS3: Payroll recovery (as of December 2021) | 2.893 | 1.02902 | 1.059 | 0.401 | −0.617 |
| Item 22 ICTS4: Still affected by the crisis | 2.8819 | 1.12253 | 1.26 | 0.504 | −0.664 |
| Item 24 ICTS5: Amount of public support | 2.6753 | 1.32696 | 1.761 | 0.096 | −1.403 |
| Item 25 ICTS6: Percentage of revenue 2020 compared to 2019 | 2.9188 | 1.1159 | 1.245 | 0.451 | −0.701 |

Note: Continuous variables were categorized using a Likert scale with values from 1 to 5.

## 4. Results

### 4.1. Exploratory Factor Analysis (EFA) and Reliability Tests

CFA methodology recommends verifying the viability of the proposed model through EFA [36]. We, therefore, confirmed the relationships among the variables—first, to determine whether the proposed model is identified and, second, to verify the factor loadings on each of the variables. Table 4 presents the results.

**Table 4.** Consistency and internal validity, average variance extracted (AVE), and composite reliability.

| Variable | Factors | Factor Loadings | | AVE | CR |
|---|---|---|---|---|---|
| Business situation | Tourism subsector (BS1) | 0.71 | 0.67 | 0.65 | 0.85 |
| | Number of workers (BS2) | 0.77 | | 0.68 | |
| | Sales volume (BS3) | −0.03 | | 0.00 | |
| | Main clients (BS4) | 0.72 | | 0.70 | |
| Organizational management | Formulation of income budget (OM1) | 0.73 | 0.86 | 0.95 | 0.75 |
| | Expenditure budget (OM2) | 1.00 | | 0.99 | |
| | Financial goals (type) (OM3) | 0.86 | | 0.85 | |
| | Cost identification (OM4) | 1.01 | | 0.95 | |
| | Productive capacity (OM5) | 0.70 | | 0.74 | |
| Innovation and development | Investment in product development (ID1) | 0.82 | 0.99 | 0.64 | 0.99 |
| | Marketing investment (ID2) | 0.64 | | 0.69 | |
| | Investment in process improvement (ID3) | 0.78 | | 0.64 | |
| | Number of years (with I + D) (ID4) | 0.80 | | 0.65 | |

**Table 4.** *Cont.*

| Variable | Factors | Factor Loadings | | AVE | CR |
|---|---|---|---|---|---|
| Impact of the COVID-19 crisis on SMEs in the tourism sector | Remote work (ICTS1) | 0.70 | 0.95 | 0.69 | 0.95 |
| | Investment in reactivation (ICTS2) | 0.84 | | 0.70 | |
| | Payroll recovery (as of December 2021) (ICTS3) | 0.93 | | 0.88 | |
| | Still affected by the crisis (ICTS4) | 0.97 | | 0.93 | |
| | Amount of public support (ICTS5) | −0.78 | | 0.61 | |
| | Percentage of revenue 2020 compared to 2019 (ICTS6) | 1.00 | | 1.00 | |

Note: Factor loadings should be above 0.7, AVE values above 0.5, and CR scores above 0.7.

The factor loadings show consistency among the factors observed and the variables, except for item 3 (sales volume), which could be collinear with other variables and whose factor loading is below the accepted minimum of 0.7 [88]. We, therefore, excluded item 3 from the model. The other items and factors show satisfactory factor loadings, demonstrating the model's internal consistency. All values over 0.5 were accepted for average variance extracted (AVE) [36] and these values range from 0.64 to 1.00. Composite reliability (CR) values are between 0.75 and 0.99—in all cases, above 0.7, indicating construct validity [82]. Next, we present the adjusted empirical model (Figure 2).

The following values were obtained in the adjusted model: Chi-square = 363, degrees of freedom =126, standardized Chi-square = 2.88, root mean square error of approximation (RMSEA) = 0.021, Tucker–Lewis index TLI = 0.951, and incremental fit index IFI = 0.960 (these two indices should be >= 0.90). The indices for goodness of fit are incremental fit CFI = 0.960 and parsimony fit NFI = 0.940. According to theory, these values confirm the model's internal consistency, causal relationships among the variables, and good fit [82].

Next, the fact that the average of our six factors is associated with ICTS indicates that tourism business owners observed a high impact from the COVID-19 crisis. The tourism SMEs adopted remote work to reduce the impact on employment in the months of lockdown but could not sustain this measure economically over time. The measurement variable public aid to maintain payroll was crucial during the most critical months.

Item 3 (sales) in 2020 compared to 2019 measured the economic impact most precisely, showing a fall in income of 50% in over half the businesses that remained active; this calculation was obtained from the original revenue figures for 2019 and 2020, extracted from Orbis.

This item affected travel agencies in lower percentages. Many business owners had to assume the cost of reinvesting to reactivate their business to adapt to changes in technology and meet public-health requirements. Finally, the findings show that trends in both sales and payroll recovery in 2021 were factors determining whether firms were still affected by the pandemic compared to the most recent year under normal circumstances (2019). This factor strengthens indications that recovery is ongoing. Table 5 presents our evaluation of each hypothesis using the statistics obtained.

**Table 5.** Hypothesis contrast with structural model results.

| Relationships | Regression Weights | Std. Dev. | *t*-Value | *p*-Value |
|---|---|---|---|---|
| The business situation of tourism SMEs in Colombia may influence the economic indicators caused by the COVID-19 crisis. | 0.815 | 0.161 | 2.815 | 0.005 |
| The COVID-19 crisis has affected the strategic management of tourism SMEs in Colombia. | 0.839 | 0.057 | 4.162 | 0.000 |
| Innovation and development in tourism SMEs have been decisive, although conditioned by the COVID-19 crisis. | 1.105 | 0.019 | 5.568 | 0.003 |

| Relationships | Regression Weights | Std. Dev. | *t*-Value | *p*-Value |
|---|---|---|---|---|
| Depending on the business situation of tourism SMEs, these firms promoted innovation and devel-opment which contributes to improving the business situation | F = −0.18 and 0.357 | 0.042 | 2.941 | 0.033 |
| Innovation and development practices support organizational management of tourism SMEs. | 4.905 | 1.635 | 4.67 | 0.004 |

Note: The regression weight obtained for the causal relationships corresponds to consolidation of the factors composing the observed endogenous variables relative to the unobserved variables.

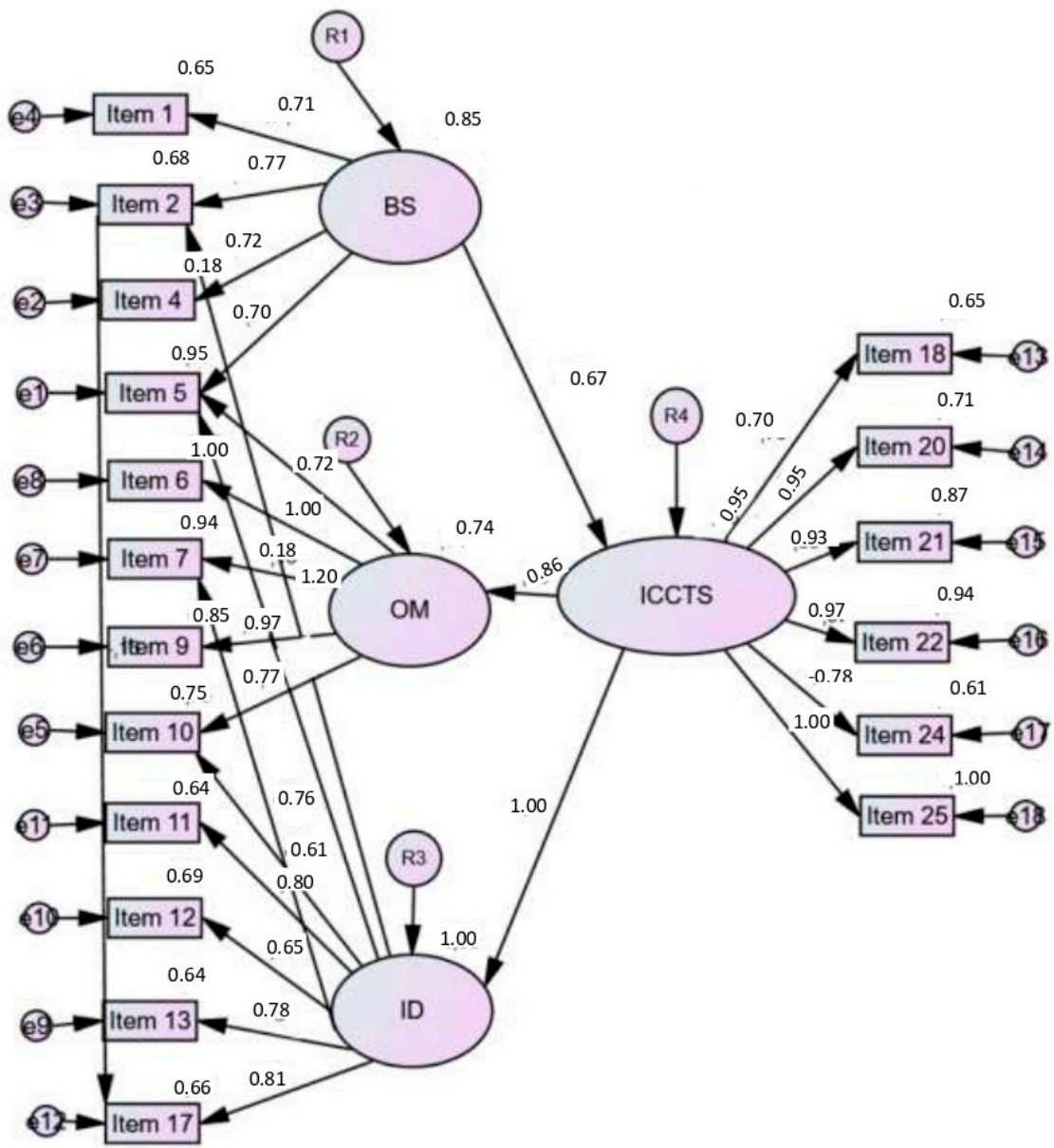

**Figure 2.** Adjusted model.

*4.2. Business Situation (BS) When Facing the Economic Impact of the COVID-19 Crisis (ICTS)*

To test the validity of hypothesis 1 about how can the business situation of tourism SMEs in Colombia influences the level of economic impact caused by the COVID-19 crisis, we considered the weight of the regression obtained, 0.815 (Table 5), and the factor

loading, 0.67. Although this factor loading is below the commonly accepted threshold of 0.7, we included it because it showed a statistically significant relationship to the criterion variable ICTS through three of the factors observed. Similarly, the t-value obtained in the hypothetical relationship is far from 0, validating the alternative hypothesis proposed, and a p-value below 0.05 permits us to reject the null hypothesis.

The economic subsector (item 1) obtained a factor loading of 0.71 (Figure 2) and a regression weight of 0.491—statistically significant values. Analysis of the percentage incomes for 2020 compared to 2019 shows that travel agencies suffered the greatest economic impact of the pandemic, with a loss of 58% income, followed by the lodging sector with 52%, and food and beverage with 40%. Tourism clubs lost 32%. We calculated these losses from the sales figures for 2019 and 2020, extracted from Orbis, classified by subsector. The percentages do not include firms that closed in 2020.

The link between number of workers (item 2) and firm size showed a factor loading of 0.77 and a regression weight of −0.676, demonstrating an inverse relationship. This finding indicates that, the larger the firm, the less severe the economic impact of the COVID-19 crisis. Sales volume (base year 2020) (item 3), did not, however, show a statistically significant factor loading. The reason may be the decrease in 2020 sales, which could exclude many companies from the SME category, as well as the incorporation into the market of new firms, for which we could not establish the degree of impact because they had no figures from previous years.

Main customers (item 4) produced a factor loading of 0.72 and a regression weight of 1.0. This result may be considered decisive for ICTs, as the statistical distribution of the variable highlights the fact that 65% of tourism SMEs sell their services primarily to consumers and families. Type of customer, thus, explains why the impact on sales was higher for these firms than for firms whose market focuses on other companies. Finally, the 11% that sell tourism services to public administrations experienced less impact from the pandemic.

### 4.3. How the COVID-19 Crisis Affected Strategic Management of Tourism SMEs

Through H2, we aimed to determine how organizational strategic management (OM) was affected by the COVID-19 crisis. The results of the CFA—factor loading 0.86, regression weight of 0.839 (Table 5)—confirm the validity of the construct proposed, indicating that organizational strategic management decreased due to the COVID-19 crisis. The following evaluates the factors composing the variable organizational strategic management.

SMEs' formulation and monitoring of the income budget obtained a statistically significant factor loading and regression weight. This item is, thus, considered significant within strategic management. For 2020, the frequency of responses showed that 52% of tourism entrepreneurs did not formulate a budget or did not follow up on proposed goals.

Cost and expenditure budget (item 6), factor loading, and regression weight were high, indicating the importance of cost management and monitoring during the crisis. Similarly, reduction in costs and expenditure was one of the crisis' most significant consequences for firms. The data obtained from the survey indicate that only 27% of business owners performed monthly or more continuous monitoring of organizational costs and expenditures.

Item 7 measures projection of financial goals and monitoring. As this item obtains a statistically low but acceptable factor loading and regression weight, we can consider it an aspect of management moderately affected by the pandemic. Frequency analysis of the responses showed that 59% of business owners formulate and monitor goals for profitability, operating margin, or all goals as a whole. Periodicity in formulation or monitoring financial goals was included in item 8 but discarded from testing of the model because it could not be identified.

Determining and monitoring fixed and variable costs (item 9) produced a statistically significant factor loading and regression weight. This finding supports the importance of management and identification for fixed and variable costs during the crisis to reduce the weight of operational leveraging. According to a previous frequency analysis (Appendix B),

54% of tourism SMEs did not identify or classify costs and this lack of management may have intensified the pandemic's impact on their firms.

Finally, for the variable OM, item 10 (productive capacity) obtained average but valid factor loading and regression weight, demonstrating that the COVID-19 crisis decreased tourism firms' productivity and caused infrastructure reductions in 44% of SMEs dedicated to tourism in Colombia.

### 4.4. Extent to Which Innovation and Development Processes (ID) Are Conditioned by the COVID-19 Crisis

H3 posed whether innovation and development in tourism SMEs were determining factors, although conditioned by the COVID-19 crisis. Factor loading and regression weight confirm this hypothesis. Both values are statistically significant, with the highest value for investment in product and service development. Further, also relevant was the number of years the firm had had an ID department. Third and fourth, tourism business owners considered investment in marketing and in process improvement as determining factors for enduring the effects of the crisis. The regression weights of these variables were high, confirming their strong influence on ID processes after the start of the pandemic. The results for recovery in 2021 indicate, however, the importance of innovating in services and marketing to achieve more efficient recovery.

### 4.5. Business Situation (BS) and Its Relationship to ID

Based on the business situation of tourism SMEs, H4 promotes ID and ID contributes to improving the business situation. Items for this hypothesis obtained statistically low factor loadings and regression weights, leading us to accept the null hypothesis and reject H4. The survey data, thus, show no significant relationship of business situation to ID processes, at least during the pandemic period.

### 4.6. How ID Influences Organizational Strategic Management

H5 proposed that ID practices support tourism SMEs' organizational management. Contrasting the hypotheses shows statistically significant factor loadings and regression weights, leading us to accept H5. The factors that support this construct indicate that ID contributes to formulation and monitoring of sales, cost, and expenditure budgets. ID, in turn, promotes achievement of financial goals and, thus, supports improvements in productive capacity in tourism SMEs.

## 5. Results Discussion

This study analyzed the economic impact of the COVID-19 crisis on the tourism sector in Colombia to explain the significance of business situation as these firms faced the crisis. The study also analyzed the implications of the pandemic for organizational strategic management and innovation and development processes. The results obtained through CFA show that the method used to analyze the data was appropriate, as it evaluated all the items and their relationships as a whole, enabling achievement of the proposed research goal. The results identify a strong impact on the economy of tourism firms, especially those with a smaller economic structure and less-developed business situations. In accordance with [10,89], smaller firms—those with fewer workers and a weaker customer base and economic and financial infrastructure—suffered more intense effects from the pandemic.

The statistical results support the theory. They indicate that both the business situation of tourism SMEs in Colombia and this situation's influence on the level of economic impact by the COVID-19 crisis required greater public assistance to firms through financing, subsidies, and job preservation, among other issues. This finding reinforces the work in [12,27]. Such aid must include accessible requirements, such as those proposed by [44,45,90], and recognize that firms have still not recovered fully from the effects of the crisis.

Like [29,30], our results show the main effect on tourism firms to occur in sales figures, which decreased, on average, 50% in 2020 compared to 2019. As a result, salaries decreased

up to 10% in some cases. The official figures also indicate that 25% of tourism SMEs closed [21]. In the tourism subsectors, economic impact was calculated relative to sales revenues (see Figure 3).

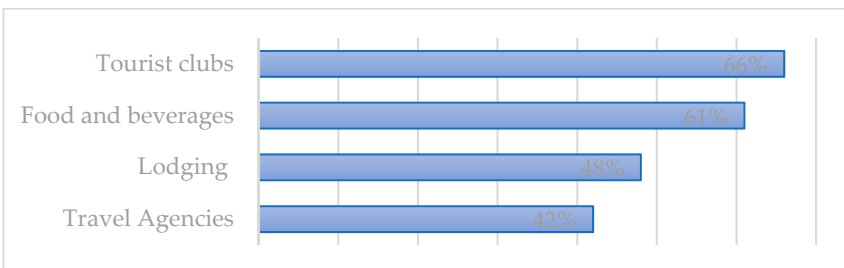

**Figure 3.** Average percentage revenue 2020/2019. Note: the figure is derived from the averages sales percentages for SMEs in each subsector in 2020 with respect to 2019. The above figure identifies the tourism subsector that experienced the largest decrease in sales during the first year of the pandemic.

Figure 3 shows that the firms most affected were travel agencies, followed by the lodging sector (a finding that coincides with [26]), and firms that sell food and beverages. Some of these results concur with [38]. The firms least affected were, thus, those whose main market is public administration and other firms, as the main customers of travel agencies and the hospitality sector are families and individuals. Hotels also showed a significant drop in sales in 2020, consistent with [18], but contradicting [85].

Analysis of the pandemic's effect on organizations' strategic management confirmed the veracity of H2; a decrease in tourism SMEs' economic capacity led to a decrease in payroll. At the same time, the companies used their productive capacity or operating leverage less, increasing the weight of fixed costs on operating results. All these conditions led to significant decreases in business infrastructure, in line with [69].

The statistical results obtained also reveal the importance of developing budgets and proposing goals and strategies. The results indicate that tourism SMEs that conducted such management experienced less negative impacts on their sales and hired personnel, as also found in [69].

ID showed clear decreases in investment, especially during suspension of economic tourism activity. These results agree with [91]. Creativity and resilience were fundamental to strategy, however, as travel resumed and borders opened, initiating progress toward recovery. These measurements concur with [15,92].

Based on the factor analysis, the results for the business situation, as measured prior to the pandemic, and for ID processes, analyzed through H4, do not appear to be significant. This finding affirms that changes in business situation during the pandemic could have led to decreases in tourism firms' ID, as stated by [16]. This finding contrasts with changes experienced during social distancing by other businesses grounded in digital resources for marketing and product and service innovation, as [93] proposes.

Finally, we analyzed the relationship between ID and strategic management through H5. The results are compatible with most current theories. For example, they align with the contribution of [94], which supports the hypothesis that innovation practices in business management and adaptability to changes contributed to firms' ability to endure the effects of the crisis, as well as to economic and financial recovery, in line with [95].

## 6. Conclusions

In pursing our goal to determine the economic–organizational impact of the COVID-19 crisis on the tourism subsectors in Colombia, this study demonstrates that the main impact on tourism firms occurred in sales figures, with an average fall of 50% in 2020 compared to 2019. Similarly, suspension of trips and mandatory distancing led to inactivity in around 60% of tourism firms, generating a decrease in payroll of up to 10% in some cases. To this serious employment situation, we must add the closing of 25% of tourism firms, reported

by official statistics. These negative effects, reflected in the national economy, are still being felt.

One measure adopted to contain the effects of the pandemic was increased remote work. This measure was useful during lockdown to reduce the impact on employment, but it was not sufficient to sustain tourism firms economically, as the reduction in revenue required them to decrease their operating costs and expenditure. Public support contributed partially to mitigating the negative effects on SMEs' sustainability, but it was not enough.

The loosening of social distancing measures and possibility of travel initiated the reactivation process in the tourism sector. However, most firms had to make significant investments to reach their customers through ICT and to fulfill the public-health requirements.

Sales for 2021 show that recovery is still underway; the number of workers in the tourism sector has not returned to 100% of pre-COVID-19 employment. In firms' internal management, the economic and financial situation gradually affected decisions about investment and management, requiring managers to prioritize essential expenses. The pandemic, thus, had serious implications for organizational strategic management and ID processes at the beginning of lockdown. This study showed that management strategies in areas related to finance and innovation facilitated tourism's gradual recovery.

Contrasting the hypotheses as a whole showed a weaker relationship between tourism SMEs' business situation when facing the pandemic crisis and investment in ID. This finding shows decreases in investment during the most critical months but also suggests that firms opting for strategic use of technology and innovation in the market and services achieved greater advances in economic recovery.

Innovation and technology, thus, support strategic management of both finance and customers. This effect occurred through formulation and monitoring of budgets for sales and costs, measures that promote achievement of financial goals and productive capacity. This activity also made it easier for tourism SMEs to reach existing and potential customers.

The study findings recommend that tourism entrepreneurs continue to strengthen management towards recovery based on strategies that integrate the economic and financial area, customers, and sales with innovation and efficient marketing, while also providing attractive, efficient, and sustainable services. The findings also suggest projecting goals and formulating strategies from all perspectives on the tourism business, based on the strengths and needs for change that the pandemic revealed. Such strategies will achieve economic and employment recovery, while also contributing to environmental and social sustainability.

Among our study's limitations is the lack of information on firms that suspended activity or ceased economic activity definitively. Another limitation is recognizable bias in some responses, which may be due primarily to the fact that the respondents were financial managers and did not cover all areas of the firm. As future lines of research, we propose more in-depth analysis of strategies for recovery in the tourism sector and replication of the model in other emerging countries.

**Author Contributions:** Conceptualization L.N.T.P., E.U.G. and E.I.C.M.; methodology, L.N.T.P., E.U.G. and E.I.C.M.; software, L.N.T.P., E.U.G. and E.I.C.M.; validation, L.N.T.P., E.U.G. and E.I.C.M.; formal analysis, L.N.T.P., E.U.G. and E.I.C.M.; investigation, L.N.T.P., E.U.G. and E.I.C.M.; visualization, L.N.T.P., E.U.G. and E.I.C.M.; supervision, L.N.T.P., E.U.G. and E.I.C.M.; project administration, L.N.T.P., E.U.G. and E.I.C.M.; funding acquisition, L.N.T.P., E.U.G. and E.I.C.M. All authors have read and agreed to the published version of the manuscript.

**Funding:** This research was funded by the Ministry of Science, Technology and Innovation, Colombia; grant by call number 860, scholarships for doctoral studies abroad 2019.

**Institutional Review Board Statement:** Not applicable.

**Informed Consent Statement:** Not applicable.

**Data Availability Statement:** Not applicable.

**Conflicts of Interest:** The authors declare no conflict of interest.

**Appendix A**

**Online Survey Form**

**General Information:**

"Email"

"Sector"

"City"

"Department"

"Company name"

"TIN (Tax identification number)"

"Role in company of person completing the survey:"

**Part I**

"1. Activity (subsector)"

1. Tourism club
2. Travel agency
3. Lodging
4. Food and beverage

"2. Number of employees:"

1. 11–50
2. 51–100
3. 101–200
4. 201–500
5. Over 500 workers

"3. Sales volume (in thousands of USD in 2020)"

1. Under 1000
2. 1000–10,000
3. 10,001–50,000
4. 50,001–100,000
5. Over 100,000

"4. Your company's sales usually come primarily from:"

1. Public administrations
2. Consumers and families
3. Other companies
4. Consumers and families and other companies
5. Other

"5. Does the company prepare a sales budget (income)?"

1. No formula—decreased greatly
2. Annual—decreased
3. Intermediate—no change
4. Monthly—increased
5. Every period-increased greatly

"6. Does the company formulate a cost and expenditure budget?"

1. No formula—decreased greatly
2. Annual—decreased
3. Intermediate—no change
4. Monthly—increased
5. Every period—increased greatly

"7. Does the company set financial goals (types of goals)"

1. None—decreased greatly
2. Growth in economic structure (active)—decreased
3. Growth and performance—no change
4. Margin and performance—increased

5.    All financial goals—increased greatly

"8. How often does your company formulate and monitor financial goals?"

1.    No formula—decreased greatly
2.    Annual—decreased
3.    Intermediate—no change
4.    Monthly—increased
5.    Every period—increased greatly

"9. Does the company identify fixed costs and production capacity?"

1.    Does not identify—decreased greatly
2.    Annual—decreased
3.    Intermediate—no change
4.    Monthly—increased
5.    Every period—increased greatly

"10. How much does the company exploit installed production capacity (percentage)?"

1.    (0–40%)—decreased greatly
2.    (41–70%)—decreased
3.    (71–90%)—no change
4.    (91–100%)—increased
5.    (>100%)—increased greatly

**Part II**

"11. Does the company invest resources in technology? Dedicated to: (Development of product/service)"

1.    (0)—decreased greatly
2.    (1–5 thousand USD)—decreased
3.    (5–20 thousand USD)—no change
4.    (20–50 thousand USD)—increased
5.    (Over 50 thousand USD)—increased greatly

"12. Does the company invest resources in technology? Dedicated to: (Marketing)"

1.    (0)—decreased greatly
2.    (1–5 thousand USD)—decreased
3.    (5–20 thousand USD)—no change
4.    (20–50 thousand USD)—increased
5.    (Over 50 thousand USD)—increased greatly

"13. Does the company invest resources in technology? Dedicated to: (Process improvement)"

1.    (0)—decreased greatly
2.    (1–5 thousand USD)—decreased
3.    (5–20 thousand USD)—no change
4.    (20–50 thousand USD)—increased
5.    (Over 50 thousand USD)—increased greatly

"14. Does the company have a department for innovation and/or marketing? Dedicated to: (Product development)"

1.    (0)—decreased greatly
2.    (1–5 thousand USD)—decreased
3.    (5–20 thousand USD)—no change
4.    (20–50 thousand USD)—increased
5.    (Over 50 thousand USD)—increased greatly

"15. Does the company have a department for innovation and/or marketing? Dedicated to: (Marketing)"

1.    (0)—decreased greatly
2.    (1–5 thousand USD)—decreased
3.    (5–20 thousand USD)—no change

4. (20–50 thousand USD)—increased
5. (Over 50 thousand USD)—increased greatly

"16. Does the company have a department for innovation and/or marketing? Dedicated to: (Process improvement)"

1. (0)—decreased greatly
2. (1–5 thousand USD)—decreased
3. (5–20 thousand USD)—no change
4. (20–50 thousand USD)—increased
5. (Over 50 thousand USD)—increased greatly

"17. How long (years) has the company had a department of innovation, research, and development?"

1. —decreased greatly
2. (1–2 years)—decreased
3. (3)—no change
4. (4)—increased
5. (5 or more years)—increased greatly

**Part III**

"18. Are your workers able to perform their functions through telework or working from home? For what percent of their paycheck?"

1. 0—decreased greatly
2. (1–30%)—decreased
3. (31–60%)—no change
4. (61–80%)—increased
5. (81–100%)—increased greatly

"19. What percentage of the extra December paycheck did the company maintain in 2020, relative to a normal month of activity?

1. (0–30%)—decreased greatly
2. (31–60%)—decreased
3. (61–80%)—no change
4. (81–100%)—increased
5. (>100%)—increased greatly

"20. If at any time the company reached 0% activity and had to resume activity, how much did it have to invest to reinitiate activity?"

1. (0)—decreased greatly
2. (1–5 thousand USD)—decreased
3. (5–10)—no change
4. (10–50)—increased
5. (Over 50 thousand USD)—increased greatly

"21. Did the company recover the extra December paycheck in 2021?"

1. Reduced it even more—decreased greatly
2. Did not decrease it further—diminished
3. Recovered 70–80% of the paycheck—no change
4. 100%—increased
5. >100%—increased greatly

"22. Do you think your company is still affected by the pandemic?"

1. Yes
2. No

"23. Has your company received some type of public aid due to the pandemic?"

1. Yes
2. No

"24. If you answered "yes" to the previous question, how much was the aid worth?"

1.  (0)—decreased greatly
2.  (1–10 million)—diminished
3.  (10–50 million)—medium
4.  (50–200 million)—increased
5.  (Over 200 million)—increased greatly

**Appendix B. Categorization of Factors**

| Variable | Factors | Measuring Ranges |
|---|---|---|
| Business situation (BS) | Tourism subsector | 1= Tourist clubs<br>2 = Travel agencies<br>3 = Lodging<br>4 = Food and beverage |
| | Number of workers | 1 = 11–50 workers—greatly reduced<br>2 = (51–100)— decreased<br>3 = (101–200)—no change<br>4 = (201–500)—increased<br>5 = (Over 500 workers)—greatly increased |
| | Sales volume (thousand USD)<br>The lower range of USD244 thousand; corresponds to the base value for SMEs in Colombia according to the sector (December 957 of 2019). | 1 = (Under 1000)—greatly reduced<br>2 = (1000–10,000)—decreased<br>3 = (10,000–50,000)—no change<br>4 = (50,000–100,000)—increased<br>5 = (Over 100,000)—greatly increased |
| | Main clients | 1 = Consumers and families<br>2 = Other companies<br>3 = Public administrations<br>4 = Others<br>5 = Consumers and families, other businesses |
| Organizational management (OM) | Formulation of income budget | 1 = Not formulated—greatly diminished<br>2 = Annual—decreased<br>3 = Intermediate—no change<br>4 = Monthly—Increased<br>5 = All periods—greatly increased |
| | Expenditure budget | 1 = Not formulated—greatly diminished<br>2 = Annual—decreased<br>3 = Intermediate—no change<br>4 = Monthly—increased<br>5 = All periods—greatly increased |
| | Financial goals (type) | 1 = None—greatly reduced<br>2 = Economic structure growth (assets)—decreased<br>3 = Growth and performance—no change<br>4 = Margin and yield—increased<br>5 = All financial goals—greatly increased |
| | Cost identification | 1 = Not formulated—greatly diminished<br>2 = Annual—decreased<br>3 = Intermediates—no change<br>4 = Monthly—increased<br>5 = All periods—greatly increased |
| | Productive capacity | 1 = (0–40%)—greatly diminished<br>2 = (41–70%)—decreased<br>3 = (71–90%)—no change<br>4 = (91–100%)—increased<br>5 = (>100%)—greatly increased |

| Variable | Factors | Measuring Ranges |
|---|---|---|
| Business situation (BS) Innovation and development (ID) | Tourism subsector Investment in product development (these values are annual) | 1= Tourist clubs<br>1 = (0)—greatly diminished<br>2 = (1–5 million)—decreased<br>3 = (5–20 million)—no change<br>4 = (20–50 million)—increased<br>5 = (Over USD50 million)—greatly increased |
| | Marketing investment | 1 = (0)—greatly diminished<br>2 = (1–5 million)—decreased<br>3 = (5–20 million)—no change<br>4 = (20–50 million)—increased<br>5 = (Greater than USD50 million)—greatly increased |
| | Investment in process improvement | 1 = (0)—greatly diminished<br>2 = (1–5 million)—decreased<br>3 = (5–20 million)—no change<br>4 = (20–50 million)—increased<br>5 = (Over USD50 million)—greatly increased |
| | Number of years (with I + D) | 1 = (0)—greatly diminished<br>2 = (1–2 years)—decreased<br>3 = (3)—no change<br>4 = (4)—increased<br>5 = (5 years or more)—greatly increased |
| Impact of the COVID-19 crisis on SMEs in the tourism sector (ICTS) | Remote work | 1 = 0—greatly diminished<br>2 = (1–30%)—decreased<br>3 = (31–60%)—no change<br>4 = (61–80%)—increased<br>5 = (81–100%)—greatly increased |
| | Investment in reactivation | 1 = (0)—greatly diminished<br>2 = (1–5 million)—decreased<br>3 = (5–20 million)—no change<br>4 = (20–50 million)—increased<br>5 = (Over USD50 million)—greatly increased |
| | Payroll recovery (as of December 2021) | 1 = It has reduced it even further—greatly diminished<br>2 = No, still diminished—decreased<br>3 = Recovered 70–80% of payroll—no change<br>4 = 100%—increased<br>5 = >100%—greatly increased |
| | Continues to be affected by the crisis Binary variable, Likert scale does not apply. | 1 = Yes<br>2 = No |
| | Amount of public support | 1 = (0)—greatly diminished<br>2 = (1—10 million)—decreased<br>3 = (10–50 million)—no change<br>4 = (50–200 million)—increased<br>5 = (Over USD 200 million)—greatly increased |
| | Percentage of revenue 2020 compared to 2019 | 1 = (<=25%)—greatly diminished<br>2 = (26–50%)—decreased<br>3 = (51–75%)—no change<br>4 = (76–100%)—increased<br>5 = (>100%)—greatly increased |

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
