# Peer review of "Economic and Organizational Impact of COVID-19 on Colombia’s Tourism Sector"

_sustainability, doi:10.3390/su142013350_

Round 1

Reviewer 1 Report

Dear Authors,

Congratulations! The paper is very relevant and represents an important contribution to better understanding the impacts of the Covid-19 pandemic, and defining appropriate strategies to face the future.

However, I think it could be improved, with a minor revision. For this review you can consider the attached file with my comments.

Best wishes.

Author Response

Thank you very much for this opportunity, we have followed your suggestions and believe the research paper has substantially improved. The changes have been done with the “track changes” mode.

Reviewer 2 Report

By its nature, the article belongs to a journal Sustainability. The topic is interesting, but not very attractive, as the article is only confirming and discussing a generally known truth. I do not recommend the article for publication in this form. The main reasons for rejection are, beyond the low attractiveness of the article:

·       Hypothesis 1: “The business situation of tourism SMEs in Colombia may influence the level of economic impact caused by the Covid-19 crisis.” cannot be verified in the proposed way. A survey among SMEs cannot confirm this hypothesis. To confirm the hypothesis, it is necessary to perform an econometric (macroeconomic) analysis.

·       Hypothesis 2: “The Covid-19 crisis has affected the strategic management of tourism SMEs in Colombia.” The hypothesis does not bring new knowledge. All companies responded to the crisis caused by the pandemic, so it is logical that they will continue to work with this risk in strategic considerations in the future.

·       Hypothesis 3: “Innovation and development in tourism SMEs have been decisive factors; but conditioned by the Covid-19 crisis.” A hypothesis formulates a generally known and verified truth. It is not clear from the hypothesis, which factors „innovation and development“, belong to. Factors of what?

·       Hypothesis 4: “According to the business situation of tourism SMEs, these firms promoted R&D and R&D contributes to improving the business situation.” Analogical notes.

·       Hypothesis 5: “I&D practices support organizational management of tourism SMEs.” What other practices should these be?

·       The authors declare that the data was collected through an online survey. What did the questionnaire questions look like? How did the individual items come from these questions? None of this is clear. There is a complete lack of explanation of the relationships between variables and factors.

·       Table 2 is graphically inappropriate, it is not clear which factors belong to the given variables (alignment).

·       It does not make sense for all items to be on a Likert scale. For example, subsector, type of financial goals, etc. In addition, others, which are usually quantitative - number of employees, costs, revenues, etc. If there was some mathematical transformation on a scale of 1-5, it is not clear how and why.

·       The proposed model is not entirely logical, or rather, it is possible, but it is not clear from the description.

·       Inconsistent line spacing and alignment of text in tables.

·       The content of Figure 3 is not clear.

·       Authors write that “Analysis of the effect on organizations’ strategic management through H2 deter-mined that the decrease in tourism SMEs’ economic capacity caused a decrease in payroll. The circumstances created by this decrease in hired personnel led to weaker focus on formulation and monitoring of budgets and financial goals in the months with the strongest restrictive measures”. This statement is precisely a demonstration that the entire model is designed poorly. If there is a rapid drop in demand (naturally or due to restrictions), then logically no sales will be generated and then logically there is no need for so many workers who become a purely cost item. In this conclusion, the reasoning is built quite the opposite. This is just an illustrative example of the proposed model being illogical, the authors mistaking causes for reactions and vice versa.

·       Linguistic processing is weak.

·       The symbol "-" indicates a range, it is better to use the symbol "," to list citations. For example, the authors mention [11] - [12] - [13], but also [14] - [16], or [12] - [73].

Author Response

(The authors gave the same response as above.)

Reviewer 3 Report

The authors presented a robust research. However, following comments would be required to be adopted to enhance the overall reach of the article.

1. English language proofreading is required for the text.

2. How were the sample size of 289 SMEs were calculated?

Was the data collected from some government agency or database? Explain Table 1.

3. The authors may present a table specifying the variables selected for the CFA along with the previous literature references. Merge with Table 2.

4. It would be better to provide the conceptual framework of this research and then the hyphothesis.

5. Explain table 5, depict the significance of  each variable.

6. Further references towards disaster recovery must be included to support the findings of this research.

7. Provide robust, point-wise recommendations in the conclusion section.

Good Luck !!

Author Response

(The authors gave the same response as above.)
